# Improved multi-site Parkinson's disease classification using neuroimaging data with counterfactual inference

**Vibujithan Vigneshwaran**[1,2]                    VIBUJITHAN.VIGNESHWA@UCALGARY.CA
**Matthias Wilms**[3,4,5]                    MATTHIAS.WILMS@UCALGARY.CA
**Milton Camacho**[1,2]                    MILTON.CAMACHOCAMACH@UCALGARY.CA
**Raissa Souza**[1,2]                    RAISSA.SOUZADEANDRAD@UCALGARY.CA
**Nils D. Forkert**[1,2,5]                    NILS.FORKERT@UCALGARY.CA

[1] *Department of Radiology, University of Calgary, Calgary, AB, Canada*

[2] *Hotchkiss Brain Institute, University of Calgary, Calgary, AB, Canada*

[3] *Department of Pediatrics, University of Calgary, Calgary, AB, Canada*

[4] *Department of Community Health Sciences, University of Calgary, Calgary, AB, Canada*

[5] *Alberta Children's Hospital Research Institute, University of Calgary, Calgary, AB, Canada*

**Editors:** Accepted for publication at MIDL 2023

## Abstract

Deep learning has led to many advances in medical image analysis for various clinical problems. However, most deep learning models are known to be sensitive to differences in the training and test data distributions, which can lead to a decrease in accuracy when applied in real-life scenarios. Thus far, various techniques have been developed to tackle this problem, primarily focusing on harmonizing feature representations from different datasets. Due to the recent increased interest in causal approaches in deep learning, explainable harmonization techniques have gained momentum lately but have not been applied broadly yet. Our study proposes a causal flow-based technique to overcome the problem of different feature distributions in multi-site data used for Parkinson's disease (PD) classification. Feature distributions from six different sites, with a total of 415 subjects (PD: 263, healthy controls: 152), were used for the experiments. A counterfactual approach to answer the question, "How would brain MRI features appear if they were obtained at a different site?" was developed using a causal normalizing flow. When tested on features from a previously unseen site, the counterfactual-based classifier demonstrated superior performance (weighted f1 = 0.68) compared to a classifier trained on purely observational data (weighted f1 = 0.36) and improved accuracy compared to a harmonization technique typically used in neurological settings (weighted f1 = 0.5). These results show that the proposed technique can effectively correct differences in multi-site feature distributions to facilitate generalizable deep-learning models.

**Keywords:** Domain shift, harmonization, normalizing flow, causality, counterfactual, Parkinson's disease

## 1. Introduction

As in many other domains, medical image analysis has benefited tremendously from the recent advancements in deep learning. However, deep learning models are known to be prone to decreased accuracy when applied to datasets with a distribution that differs from datasets used for training (Robinson et al., 2020). This phenomenon, known as domain shift, can

be caused by acquisition shifts, population shifts, or a combination thereof. An acquisition shift occurs when there are variations regarding the imaging modalities, procedures, or scanners used, while a population shift can occur when the subject groups have different characteristics, such as demographics or disease states. Both shifts may also occur in multi-center studies making it often difficult to train generalizable machine learning models. Addressing these domain shifts is critical for improving performance across multiple sites, avoiding biases in the results, and developing generalizable deep learning models for health care.

One way to overcome the problems of domain shifts is to collect and train the models with large datasets. However, this approach has several limitations. First, collecting diverse and unbiased data can be time-consuming and labeling the data can be laborious. Additionally, training a single model to perform well across multiple sites with varying domain shifts may result in a trade-off between accuracy and generalization. Thus, it may be necessary to re-train deep learning models with new data to ensure that they perform well using data from new sites. As an alternative, researchers have explored the use of domain adaptation techniques to overcome these problems (Farahani et al., 2020), which aim to learn domain-invariant feature representations.

Domain adaptation in machine learning often involves, but is not limited to, disentangling domain-specific features from datasets or enforcing a specific distribution on the latent representations of the inputs. Typically, ComBat (Johnson et al., 2007) and its variants are used for this purpose. This empirical Bayes method aims to remove effects specific to a particular scanner or site while maintaining the relevant features of the data. A potential drawback of this approach is that it may also remove unknown confounders during the site adaptation process, decreasing the model's overall accuracy. More recently, deep generative networks, including generative adversarial networks (GANs) and variational autoencoders (VAEs), have been used for this purpose due to their ability to reconstruct images from different domains (Zhu et al., 2017; Mathieu et al., 2019). However, a major limitation of these generative approaches is their lack of interpretability and explainability, as they operate mostly as black boxes (Vasudevan et al., 2021).

Recently, causal approaches have started gaining popularity in the deep learning community to overcome the problems associated with black box machine learning models. The explicit modeling of causal relationships in deep learning settings makes the models more explainable. At the same time, it can address the issues of learning spurious correlation during training and being sensitive to domain-specific statistics. For this reason, recent studies have also started integrating causality for disentanglement tasks (Goudet et al., 2018; Bengio et al., 2019; Parascandolo et al., 2018; Yang et al., 2021). In the context of our study, causal analysis can be used to remove the effects of a particular site/scanner from the feature distributions using counterfactuals. The counterfactual inference is essentially asking the query from the trained causal model, "How would the features look like if they had been acquired at a different site?". This query synthesizes a new set of features with similar distribution characteristics from the specified site, thereby minimizing domain shifts. Thus far, such queries have been limited to a few variables, and to accurately perform them on high-dimensional data, deep invertible models that explicitly map densities are required.

Normalizing flows (NF) are a class of deep generative models that explicitly model complex data distributions (Kobyzev et al., 2021). This is achieved by representing the density

of the data as an invertible transformation of a noise variable with a simple distribution using the change of variable formula. Pawlowski et al. (2020) have developed a deep structural causal model combining causality, NF, and VAEs. This causal model was able to generate realistic counterfactual images using the Morpho-MNIST and brain MRI datasets. In a subsequent study, Wang et al. (2021) applied this framework for harmonizing multi-site image features. However, this framework requires separate conditional density estimation (Trippe and Turner, 2018) to encode the causal structure in the data. As an alternative approach, Wehenkel and Louppe (2020) argued that NF on itself could be understood as a causal model if the order of conditioners (autoregressive or coupling) was defined correctly. Following this, Khemakhem et al. (2021) proved that causal structural equation models (SEM) could be modeled using standard NFs and developed the causal autoregressive flow (CAREFL) framework. They generated counterfactual queries using a synthetic dataset of two cause and two effect variables and validated them.

This study builds upon the CAREFL framework to create counterfactual features from multi-site brain MRI data. Specifically, the main contributions of this work are (1) extending the CAREFL framework's coupling conditioner to work with different sizes of cause and effect variables and (2) for the first time, using a causal NF without external conditioning for data harmonization. Results from this study show that causal flow-based techniques could effectively correct differences in feature distributions, leading to generalizable deep learning models.

## 2. Background

### 2.1. Structural equation modeling

Structural equation modeling (SEM) is a well-known method for analyzing relationships between variables, both observed and latent, by creating a model that represents the relationships as equations. Let's assume that the feature vector $\mathbf{x}$ is d-dimensional and consists of features $\mathbf{x} = (x_1, x_2, ..., x_d)$ with a joint probability distribution $P_x$. The structural equation for each variable $x_j$ is defined as $S_j : x_j = f_j(pa_j, n_j)$, where $pa_j$ denotes the parent of $x_j$ in the causal model and $n_j$ represents the mutually independent exogenous noise variable of the noise distribution $P_n$. Then, a SEM is defined as $(S, P_n)$, where $S = \{S_1, S_2, ..., S_j\}$. In this formulation, the observational distribution of variables ($P_x$) can be thought of as being generated by sampling from a noise distribution ($P_n$) and then applying a set of structural equations ($S$) to the sampled values. This means that the observed variables are influenced by the noise distribution as well as the relationships represented by the structural equations. A directed acyclic graph (DAG), $\mathcal{G}$, either discovered or provided, is typically used to define the causal relationship between the variables. In $\mathcal{G}$, each node corresponds to a variable $x_j$, and the directed edges denote the causal ordering of the variables. In this context, structural equations can be rewritten as

$$x_j = f_j(x < \pi(j), n_j) \tag{1}$$

where $x < \pi(j)$ represents all variables before $x_j$ in the causal ordering defined by the graph.

### 2.2. SEM with Normalizing flows

Normalizing flows model a complex probability distribution $p_x$ as the result of a series of transformations $\mathbf{T}$ applied to a predefined probability density $p_z$, typically chosen as Gaussian (Kobyzev et al., 2021). $\mathbf{T}$ has to be invertible and differentiable to train the model using the change of variables formula

$$P_x(\mathbf{x}) = P_z(\mathbf{T}^{-1}(\mathbf{x}))|\det J_{\mathbf{T}^{-1}}(\mathbf{x})| \tag{2}$$

where $\det J_{\mathbf{T}^{-1}}$ is the determinant of Jacobian $J_{\mathbf{T}^{-1}}$. The series of transformations $T = T_1 \circ \cdots \circ T_k$ is typically implemented using neural networks while conditioning the Jacobian as a lower triangular matrix for efficient computation of the determinant. Coupling conditioners are typically used in normalizing flows to yield a lower triangular $J$. The conditioners $c_j$ for each variable $x_j$ are defined as:

$$c_j = \begin{cases} \underline{h}_j & \text{if } j \leq k \\ h_j([x_1 \cdots x_k]) & \text{if } j > k \end{cases} \tag{3}$$

where $\underline{h}_j$ denotes constant values and $h_j$ is a function of previous $x$ values, and $k \in [1, d]$ is a hyperparamter. Thus, each transformed input variable from the corresponding latent variable $z_j$ is given by:

$$x_j = T_j(z_j, c_j) \tag{4}$$

One might notice the similarity between the equations 1 and 4. More precisely, both models define a specific order for variables and assume that the latent variables $z_j$ follow simple distributions. This ordering can be determined from a causal DAG using D-separation, which reveals conditional independence in the graph. The conditioning $c_j$ for each variable $x_j$ in normalizing flows can be set based on this identified conditional independence, and this ordering of the conditioning must be maintained throughout the flow. For more information on this topic, the reader can refer to the works of Wehenkel and Louppe (2021) and Khemakhem et al. (2021).

### 2.3. Counterfactual inference

Counterfactual queries aim to assess statements about hypothetical situations. For example, if variable $x_j$ had taken the value $x_j = \alpha$ in our observed feature vector $\mathbf{x}^{obs}$, what would be the value of variable $x_i$? This is denoted as $x_{i,x_j \leftarrow \alpha}$. According to Pearl (2009), evaluating causal counterfactuals requires three steps: abduction, action, and prediction. After fitting an NF to the data, abduction evaluates the posterior distribution over latent variables $\mathbf{z}^{obs}$ given observations $\mathbf{x}^{obs}$. In normalizing flow models, this can be simply done by computing the transformation $\mathbf{z}^{obs} = \mathbf{T}^{-1}(\mathbf{x}^{obs})$ to identify the corresponding latent values, where $\mathbf{z}^{obs}$ is the latent representation. The next step is to intervene and fix the value of $x_j$ to a specific value $\alpha$, which makes it independent of its causes $pa_j$ and noise $n_j$. This is referred to as the action step and is denoted by $do(x_j = \alpha)$. In this step, the corresponding value of change of $x_{i,x_j \leftarrow \alpha}$ is adjusted in the latent space $\mathbf{z}_j^{obs}$. After intervening, the new feature vector $\mathbf{x}^{cf}$ is predicted by computing a transformation pass of the intervened $\mathbf{z}^{obs}$ in the NF model.

## 3. Material and Methods

### 3.1. Data

We utilized data from 6 sites, including 415 subjects (263 with Parkinson's disease and 152 healthy) to develop and evaluate our methods. The studies included were the BioCog (Clinical, Magnetic Resonance, and Genetic Biomarkers of Cognitive Decline and Dementia in Parkinson's Disease) (Acharya et al., 2007), C-BIG (Montreal Neurological Institute's Open Science Clinical Biological Imaging and Genetic Repository), Neurocon (Badea et al., 2017), Tao Wu (Badea et al., 2017), OpenNeuro Japan(Noritaka et al., 2018), and PD MCI Calgary (Lang et al., 2019). For this work, we included subjects with T1-weighted magnetic resonance imaging (MRI) datasets with known ground truth labels (PD or healthy controls). All studies included in this work received ethics approval from their local ethics boards and written informed consent from all the participants in accordance with the Declaration of Helsinki.

In the initial step, the T1-weighted images were pre-processed to remove non-brain tissue using HD-BET (Isensee et al., 2019), resampled to 1mm isotropic resolution employing a linear interpolation, and corrected for bias field distortions (Tustison et al., 2010). Afterwards, each T1-weighted MRI dataset was registered non-linearly to the MNI PD25 atlas. The Harvard-Oxford (HO) cortical and subcortical segmentations available in the atlas space were then transformed inversely using nearest-neighbor interpolation for each subject. The HO atlas contained volumes of 69 brain regions, including the hippocampus, caudate, and insular cortex, which were calculated from the transformed atlas and normalized based on the intracranial volume. The final dataset comprised 70 features, including the intracranial volume. Figure 1(left) displays the distribution of the normalized right hippocampus volume for each site. It can be observed that there are inter-site variations in the distribution, which could potentially be attributed to population and acquisition shifts.

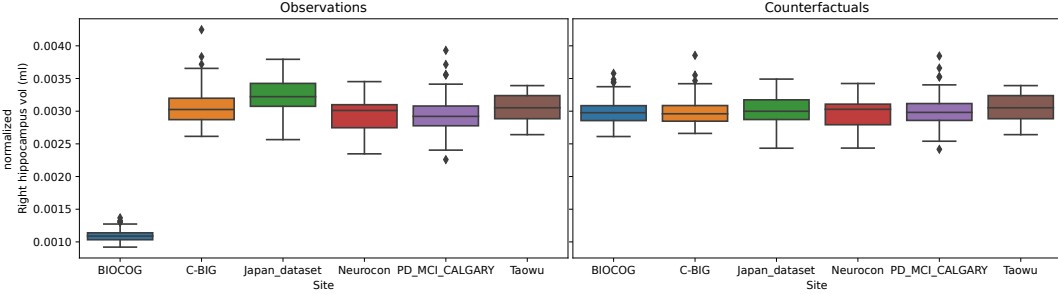

Figure 1: The figure shows the multi-site distribution of the observed right hippocampus volume on the left and the inferred counterfactual distribution of the same features on the right (base site for counterfactual: PD MCI Calgary).

### 3.2. Counterfactual inference of sites

In the first step, a causal DAG was defined with the assumption that the site variable influences every feature and that there is no interaction between features. Then, we asked the counterfactual query, "How would the features look if they had been acquired at a different site?". By following the abduction, action, and prediction steps mentioned in section 2.3, counterfactual features were generated with respect to the base site. Specifically, the site variable $s_i$ was set to the base site b as $do(s_i = b)$. Then, the flow was computed to generate the counterfactual features.

A coupling conditioner was designed to encode this DAG into the normalizing flow architecture (see equation 3), and the order of the coupling was maintained throughout the layers. The normalizing flow model consisted of 10 transformation layers similar to realnvp (real-valued non-volume preserving) layers (Dinh et al., 2017). A multi-layer perceptron with three hidden layers, each layer having ten neurons, was used to determine the transformation parameters. As usual for a bijective flow model, the layer's size was the input feature size, which was consistent for every layer. All input features were normalized during training using the mean and the standard deviation. For learning, the Adam optimizer was used with a learning rate of 0.0005 and a batch size of 128. A negative log-likelihood estimator determined the loss of the training, where the standard normal distribution was used as the latent distribution.

### 3.3. Classification network

In order to evaluate the benefit of using the proposed data harmonization technique, a simple PD classification model was developed and trained using the raw as well as corrected volume features. A multi-layer perceptron with three hidden layers was used for this purpose. The input layer size was 70, equal to the number of brain MRI features extracted. The subsequent hidden layers had 64, 32, and 16 neurons, batch normalization, and ReLU activations. The output layer employed sigmoid activation, and binary cross-entropy was used as the loss function. Since the number of PD versus healthy subjects was not balanced, we used a weighted random sampler with replacement for the batching. For optimization, Adam was used with the following parameters: learning rate = 0.005, weight decay = 0.0001, and batch size = 64. During training, the validation accuracy was tracked, and the model with the best validation accuracy was chosen, whereas twenty percent of the training data was used for the validation. PyTorch was used for the implementation, and the model was trained using a 32GB NVIDIA 3090 graphics card.

### 3.4. Experimental setup

Following the feature extraction, the final dataset consisted of the study name ($\mathbf{s}$), 70 brain MRI features (X), and a label for disease classification ($\mathbf{y}$). Next, the datasets were split as 60% for the training and 40% for the testing. The ratio of PD vs healthy in the dataset was maintained as the same in training and testing datasets through stratification. For the experiments, studies were divided into three sets: base site (B), existing sites (E), and new site (N). The new site N was selected to be the site with drastically different characteristics and used to simulate the scenario in that the model is used to classify unseen data. More

precisely, N was chosen to be BioCog, and one site from {Neurocon, Japan, PD MCI Calgary, C-BIG, Tao Wu} was selected as B and the rest as E.

In the first step, a normalizing flow model was trained separately for each site, with site B as the base. For each site $e$, features $X_e^{\text{train}}$ and $X_B^{\text{train}}$ were used for this purpose. After training, counterfactual data was generated for both train and test data from each site $e$ and denoted as $X_{e \leftarrow B}^{\text{train}}$ and $X_{e \leftarrow B}^{\text{test}}$. These steps are schematically shown in Fig. 2, and the inferred counterfactual features for each site are shown in Fig. 1(right). It is important to note that **y** was not used in the NF models to simulate unlabeled data and to avoid labels leaking into the flow models during testing.

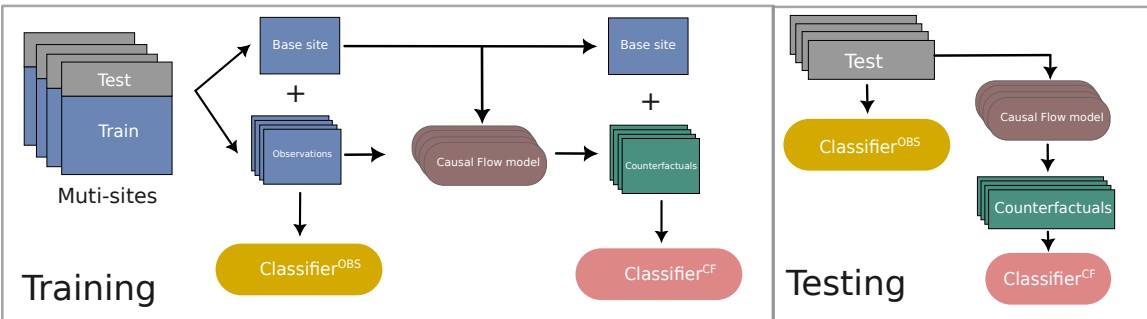

Figure 2: The training and testing process of the proposed framework. Training dataset was used to train flow and classifier models, and F1-score was calculated on the test dataset.

To evaluate the benefit of the proposed data harmonization method, we trained three classifiers, $C_{\text{obs}}$, $C_{\text{ComBat}}$, and $C_{\text{cf}}$ with the same hyper-parameters, one with true observational data, one with ComBat harmonization, and the other with counterfactuals. ComBat (Johnson et al., 2007) is a data harmonization technique typically used in neurological settings to harmonize multisite image-derived features. The open-sourced python code from the authors was used for the implementation of ComBat harmonization (Fortin et al., 2018).

As the next step, $D_{\text{obs}}^{\text{train}}$ $(X_{E+B}^{\text{train}}, \mathbf{y}_{E+B}^{\text{train}})$ was used for the training of $C_{\text{obs}}$ and ComBat harmonized data $D_{\text{ComBat}}^{\text{train}}$ for $C_{ComBat}$. Similarly for $C_{\text{cf}}$, training dataset $D_{\text{cf}}^{\text{train}}$ ( $X_{(E \leftarrow B)+B}^{\text{train}}, \mathbf{y}_{E+B}^{\text{train}})$ was used. Final testing was done using $D_{\text{obs}}^{\text{test}}$, $D_{\text{ComBat}}^{\text{test}}$ and $D_{\text{cf}}^{\text{test}}$ on $C_{\text{obs}}$, $C_{\text{ComBat}}$, and $C_{\text{cf}}$, respectively. Next, it was tested how the model behaved when it was presented with unseen data. Using the trained models $C_{\text{obs}}$, $C_{\text{ComBat}}$ and $C_{\text{cf}}$, test data from the new site, respective ComBat harmonized data, and counterfactual data were tested respectively.

## 4. Results

First, the accuracy of the NF models was validated by comparing the predicted and observational distributions (Appendix A). Then, PD classification accuracy was measured on test datasets using both predicted and observational features. For this, separate classifiers were trained with these 70 features as inputs. The test results of $C_{\text{obs}}$, $C_{\text{ComBat}}$, and $C_{\text{cf}}$

(base: C-BIG) are shown in Fig. 3, which displays the improvement in the overall test accuracy after making the counterfactual inference. Specifically, the PD MCI CALGARY site showed significant improvement in accuracy compared to observational and ComBat harmonized data.

When unseen data was tested with the classifiers, the counterfactual-based classifier showed substantially better performance (weighted f1 = 0.68) compared to a classifier trained on purely observational data (weighted f1 = 0.36) and ComBat-harmonized data (weighted f1 = 0.50). It can be noted that before counterfactual inference, all observational data from the unseen site was classified as healthy because of the difference in the distributions. The same experiment was conducted with different base sites, and their accuracy is presented in Table 1, where precision and recall results are presented in Appendix B. Similar test accuracies were observed for classifiers with different base sites, showing that the choice of the base site does not affect the classification.

Table 1: Weighted F1-scores of classifiers trained with different base sites. CF: Causal flow based technique, PMC: PD MCI CALGARY

| Base site | Technique | C-BIG | Japan | Neurocon | Taowu | PMC | BIOCOG |
|-----------|-----------|-------|-------|----------|-------|------|--------|
| - | - | 0.67 | 0.56 | 0.48 | 0.66 | 0.59 | 0.36 |
| C-BIG | CF | 0.65 | 0.62 | 0.5 | 0.6 | 0.64 | 0.68 |
| | ComBat | 0.66 | 0.6 | 0.48 | 0.63 | 0.46 | 0.5 |
| Japan | CF | 0.65 | 0.62 | 0.5 | 0.6 | 0.64 | 0.74 |
| | ComBat | 0.66 | 0.6 | 0.48 | 0.63 | 0.46 | 0.5 |
| Neurocon | CF | 0.67 | 0.62 | 0.5 | 0.53 | 0.66 | 0.7 |
| | ComBat | 0.66 | 0.6 | 0.48 | 0.63 | 0.46 | 0.5 |
| Taowu | CF | 0.67 | 0.62 | 0.5 | 0.6 | 0.64 | 0.71 |
| | ComBat | 0.66 | 0.6 | 0.48 | 0.63 | 0.46 | 0.5 |
| PMC | CF | 0.67 | 0.62 | 0.5 | 0.6 | 0.61 | 0.68 |
| | ComBat | 0.66 | 0.6 | 0.48 | 0.63 | 0.46 | 0.5 |

Furthermore, we analyzed which features are predominantly affected by the proposed harmonization technique. The mean squared error (MSE) was calculated by comparing the observational and counterfactual data from the BIOCOG site. The five most affected features and their respective MSE values are as follows: left cerebral cortex=0.0279, right cerebral cortex = 0.0276, right cerebral white matter=0.0067, left cerebral white matter=0.0066, frontal pole=0.0014.

## 5. Conclusion

This study investigates causal counterfactual inference as an approach to data harmonization to address differences in feature distributions in multi-site data used for Parkinson's disease (PD) classification. Unlike other generative techniques, the causal normalizing flow method explicitly models the relationships between variables based on a causal structure with a tractable density, enabling direct counterfactual inference. Furthermore, in contrast

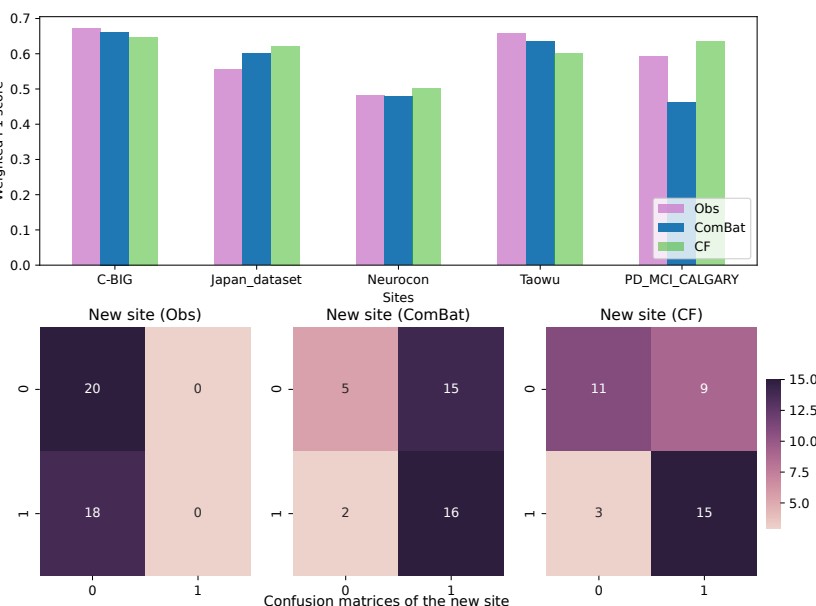

Figure 3: Top: PD classification accuracy of true observed data (Obs), ComBat harmonized data (ComBat), and counterfactuals (CF) using C-BIG as the base site. Bottom: test results for an unseen site represented by the confusion matrix.

to other flow-based harmonization methods, this proposed technique facilitates the construction of causal graphs without external conditioning. Using this approach, we sought to answer the question, "How would brain MRI features appear if they were obtained from a different site?".

The present investigation demonstrates promising outcomes in terms of classification accuracy by leveraging image-derived features. The inferred counterfactual-based classifier's overall accuracy was better than that of a classifier trained on purely observational data. When tested on features from a previously unseen site, the counterfactual-based classifier demonstrated improved performance compared to ComBat, a technique typically used in neurological settings. These results indicate that the proposed technique can effectively harmonize the multi-site distributions to facilitate generalizable deep learning models. One of the primary limitations of this study is the constrained dataset employed. However, as a future direction, our methodology can be extended to incorporate large image datasets. Furthermore, this technique can easily be extended to other relevant areas, such as fairness and bias mitigation in deep learning.

## Acknowledgments

This work was supported by the Canada Research Chairs program and the River Fund at Calgary Foundation.

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

## Appendix A

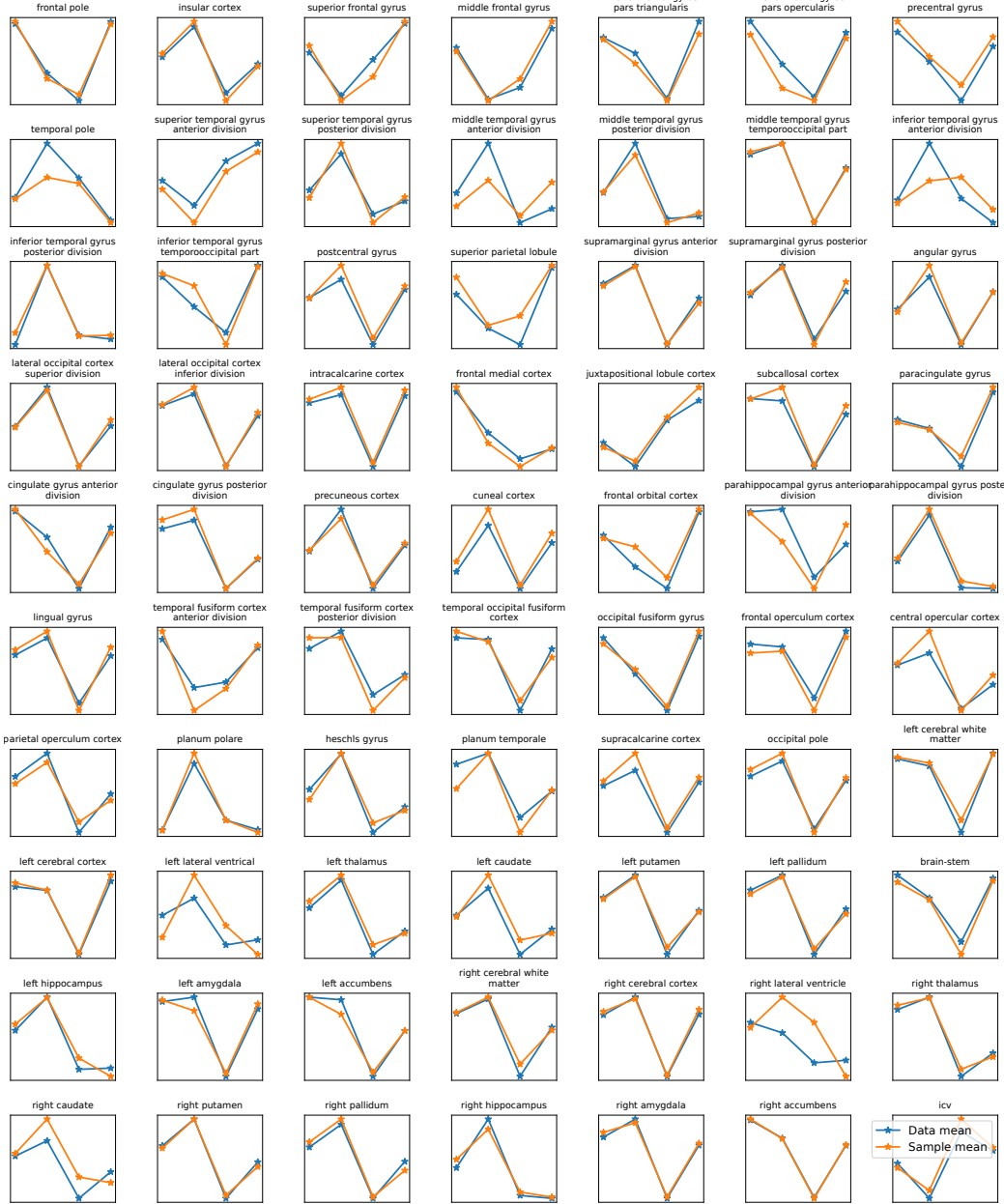

Figure 4: Blue: Mean values of the observational dataset features. Orange: the respective expected value of the learned probability density function using normalizing flows.

## Appendix B

Table 2: Precision of classifiers trained with different base sites

| Base site | Technique | C-BIG | JD | NC | Taowu | PMC | BC |
|---|---|---|---|---|---|---|---|
| - | - | 0.76 | 0.56 | 0.52 | 0.71 | 0.59 | 0.28 |
| C-BIG | CF | 0.75 | 0.67 | 0.5 | 0.6 | 0.63 | 0.71 |
| | ComBat | 0.72 | 0.59 | 0.4 | 0.81 | 0.43 | 0.62 |
| JD | CF | 0.75 | 0.67 | 0.49 | 0.6 | 0.63 | 0.75 |
| | ComBat | 0.72 | 0.59 | 0.4 | 0.81 | 0.43 | 0.62 |
| NC | CF | 0.76 | 0.67 | 0.49 | 0.54 | 0.66 | 0.75 |
| | ComBat | 0.72 | 0.59 | 0.4 | 0.81 | 0.43 | 0.62 |
| Taowu | CF | 0.76 | 0.67 | 0.49 | 0.6 | 0.63 | 0.73 |
| | ComBat | 0.72 | 0.59 | 0.4 | 0.81 | 0.43 | 0.62 |
| PMC | CF | 0.76 | 0.67 | 0.49 | 0.6 | 0.6 | 0.71 |
| | ComBat | 0.72 | 0.59 | 0.4 | 0.81 | 0.43 | 0.62 |

Table 3: Recall of classifiers trained with different base sites

| Base site | Technique | C-BIG | JD | NC | Taowu | PMC | BC |
|---|---|---|---|---|---|---|---|
| - | - | 0.61 | 0.56 | 0.47 | 0.67 | 0.61 | 0.53 |
| C-BIG | CF | 0.58 | 0.61 | 0.53 | 0.6 | 0.65 | 0.68 |
| | ComBat | 0.61 | 0.61 | 0.59 | 0.67 | 0.51 | 0.55 |
| JD | CF | 0.58 | 0.61 | 0.53 | 0.6 | 0.65 | 0.74 |
| | ComBat | 0.61 | 0.61 | 0.59 | 0.67 | 0.51 | 0.55 |
| NC | CF | 0.61 | 0.61 | 0.53 | 0.53 | 0.67 | 0.71 |
| | ComBat | 0.61 | 0.61 | 0.59 | 0.67 | 0.51 | 0.55 |
| Taowu | CF | 0.61 | 0.61 | 0.53 | 0.6 | 0.65 | 0.71 |
| | ComBat | 0.61 | 0.61 | 0.59 | 0.67 | 0.51 | 0.55 |
| PMC | CF | 0.61 | 0.61 | 0.53 | 0.6 | 0.63 | 0.68 |
| | ComBat | 0.61 | 0.61 | 0.59 | 0.67 | 0.51 | 0.55 |

