# OpenReview forum: "Improved multi-site Parkinson's disease classification using neuroimaging data with counterfactual inference"
_MIDL.io/2023/Conference — MIDL 2023 Poster_

### Official Review · Reviewer_FkoX · 2023-02-04

**Confidence:** 4
**Preliminary Rating:** 3
**Recommendation:** Oral, Poster

**Summary:**

The problem addressed is distribution shift when DL is deployed in a clinical setting. This is an important problem.

The proposed approach is to remove site effect from DL model for brain analysis from MRI. The model used is a Normalising Flow. They build on the CAREFL model to avoid the explicit description of external conditions.


**Strengths:**

The introduction is very good at summarizing the problem and the main approaches.

The paper is clear and well written, although some important details are left out.

Sound method and experimental method


**Weaknesses:**

The dataset is relatively small (415 subjects) given that 6 sites are studied. However, bigger size are challenging to obtain. As a case study, using very large public datasets would be better to demonstrate the method value (e.g. ENIGMA, ADNI,…).

The evaluation could be improved (see detailed comments)

The method presented is not specific to Medical Imaging since the input data are tabular volume values of brain parcellation, it pertains more to biostatistics.

**Deanonymize Review:**

no

**Detailed Comments:**

Using F1-score is not ideal for clinical data. Metrics should be depending on the clinical problem. Here, I assume, that it is about Diagnosis, is that true? Specificity for fixed sensitivity might be better, and/or AUC. In the current comparison, same F1 score could be achieved with 2 different operating points, and a lower F1-score might be better for a clinical task. In that sense the Figure 3 is much more informative than Table 1, although they relate to site not the Dx GT.

Using the same classifier, what is the Dx classification accuracy with all the sites mixed together and an extra input coding the site (e.g. hot vector)? This could be one baseline.

**Paper Type:**

methodological development

**Questions To Address In The Rebuttal:**

Please describe in the text what is actually calculated on each of the 69 volume (volume and how?, just regular average?).

The paper mentions normalisation of volume with ICV, but the figure 1 shows absolute hippocampus volume. Is it normalised or not? This is especially unclear that in section 3.3 there is a comparison of absolute and corrected volume: was it corrected with ICV or corrected by the proposed method? Ideally the comparison should be 3-fold: absolute, ICV-corrected, Model-corrected.

Classifiers should be better defined in the text (site or Dx?).

The paper needs to list more clearly what are the inputs/outputs and the size of the features/conditions for each layer.

Can more clinically meaningful metrics be used instead of F1?

---

### Official Review · Reviewer_nA2p · 2023-02-05

**Confidence:** 4
**Preliminary Rating:** 2

**Summary:**

This paper offers a solution for imaging data harmonization leveraging a causal flow-based technique. The technique is validated on imaging data in Parkinson's disease from 6 different sites. The proposed method shows higher predictive accuracy for the multi-site data compared to a model trained purely on observational data.

**Strengths:**

The problem of data harmonization in imaging datasets collected across multiple sites or scanners is significant and especially relevant given the rising need to combine datasets to train more powerful deep learning models. From the methodological perspective, the paper has potential, especially in bridging between causal modeling and image data harmonization.

**Weaknesses:**

- No proper comparison is provided to other well-used data harmonization methods. The paper could be significantly improved by proper comparison to other state-of-the-art harmonization techniques. There are many methods, but some examples that come to mind are MISPEL (Torbati 2022) and Intensity warping (Wrobel 2020).
- The methods section is not well-written and especially does not provide an in-depth rationale for using statistical techniques used and has limited discussion on how concepts in causal modeling relate to the harmonization problem under study.
- While the dataset is very complex, the paper does not provide a detailed evaluation of the proposed method beyond the F1 score and confusion matrix. For example, the paper could be improved by providing examples of visualization for imaging data that are most affected by the harmonization pipeline and providing more details on how the harmonization is contributing to improved accuracy.
- The statistical analyses do not contain confidence intervals and are hard to interpret.




**Deanonymize Review:**

no

**Paper Type:**

methodological development

**Questions To Address In The Rebuttal:**

The authors could significantly improve the paper by proper comparison to other state-of-the-art harmonization techniques. The methods section could be improved by providing more context on how the proposed statistical framework synergizes with the imaging data question under study. A more detailed assessment of the model evaluation could improve the results section.

---

### Official Review · Reviewer_jpfE · 2023-02-10

**Confidence:** 4
**Preliminary Rating:** 2
**Recommendation:** Poster

**Summary:**

In this manuscript, the authors propose to use counterfactual inference and normalizing flows to
harmonize brain MRI-derived features from multiple sites. They show improvement in Parkinson’s
disease (PD) classification after their harmonization. In their comparison, they perform disease classification in original and harmonized data.

**Strengths:**

The work is of high interest in the neuroimaging
community, as multi-site data analysis is becoming more and more important in clinical studies.
However, there are some weaknesses that are listed as follows.

**Weaknesses:**

The proposed model assumes a causal relationship between the site and the MRI features. Although this makes sense for acquisition shifts, it may not be true for population shifts (see definitions in Section 1). Being from a certain demographics (e.g., a geographical region) may simultaneously both influence the subject to be part of a certain study site and affect the MRI features, creating a noncausal relationship between the site and the MRI features. This has not been discussed in the paper.

The proposed harmonization approach has not been compared to any other harmonization methods, weakening the validation.

Section 3.4 mentions: “It is important to note that y was not used in the NF models to simulate unlabeled data and to avoid labels leaking into the flow models during testing.” But it also mentions that “data from each study was split into train and test datasets with a 60:40 proportion stratified by y”. Does this mean that each training and testing fold had exactly a 60:40 ratio of PD and control subjects? If datasets were sampled by taking the disease classification (y) into account, then the models were not created by totally ignoring y. In particular, the issue of different portions of the PD patients in different sites (disease-related population shift) seems to have been manually removed here before the experiments, relieving the models from having to learn it from the data.

Although the authors show a decrease in inter-class variability after harmonization (Figure 1), this only demonstrates an increase in precision and reliability of the features, without assessing how the accuracy of the features change (for better or worse). In other words, is the volume of the right hippocampus more accurate after harmonization? For instance, a toy “harmonization” method can map all the features to a constant, thereby creating zero inter-class variability, while completely compromising accuracy. The authors, however, show an improvement in predicting disease status, which serves as a surrogate measure of usefulness of their harmonization method (but still not accuracy of the features).

Although the results are generally favorable, it appears that harmonization deteriorated the disease classification accuracy of the “C-BIG” and “PD MCI Calgary” sites, when the site itself was used as the Base, which is quite counterintuitive to me. I would expect that choosing a site as the Base should not change the results at least for that very site. Is there an explanation for this behavior?

The novelty of the work is moderate at best, given that it is heavily based on prior work that the authors correctly cite in sections 1 and 2.

In Section 3.3, it is mentioned that “The model with the best validation accuracy was chosen”. It is not clear to me what different models the authors compared here. If the validation results were used to choose a model among others and report the results for only that chosen model, then the ground truth in the validation set has been used in validation itself, which would bias the evaluation towards overestimating the classification accuracy.


**Deanonymize Review:**

no

**Paper Type:**

methodological development

**Questions To Address In The Rebuttal:**

It would be nice to list the 70 brain MRI features in the titles of the Figure 4 subplots. Also, what is the x-axis in that figure? (There are too few data points for it to be the site.)

Please see the rest of the weaknesses above.

---

### Meta-Review · Area_Chair_hTbj · 2023-02-26

**Recommendation:** Accept (Poster)
**Confidence:** 3

**Metareview:**

* reviewer jpfE  contacted me after the review-edit-deadline that they would like to improve their score to a Weak Accept following the rebuttle.

The authors propose a method the builds on counterfactual inference and deep normalizing flows to harmonize neuroimaging features across different sites. This paper is, overall, very borderline for MIDL. The original reviews were not too positive, but given their sufficient rigor, there was plenty of discussion to be had. I am slightly leaning to acceptance because the discussion/rebuttal period seems to have fixed some of the concerns of the reviewers.

In particular, the authors have made improvements to their paper, clarifying aspects about the method and evaluation, and adding metrics and (precision, recall, visualizations) experiments (comparison to e.g. combat). This is how the process is supposed to work, and I believe the reviewers' and authors' efforts have led to the improvement of the paper.

There are several concerns that I think are still standing. The amount of data is fairly limited considering the number of sites, and it is hard to assess the significance of the results. However, I think that this is a harsh reason not to accept a paper, if the method seems reasonable. Nevertheless, this probably warrants a small discussion in the paper. I am also disappointed that the comment about statistical confidence intervals was not addressed. One remaining concern is that the paper is really only dealing with image-derived features, rather than images, which makes it slightly(?) out of scope for MIDL. It would be nice if the authors assessed the part of the pipeline that goes from images to features, and see how the variance there affects confidence intervals in their results, for example.

Overall, I think the authors should address the remaining concerns in at least a one-paragraph discussion on shortcomings. As I said, it is a borderline paper, much improved from the original submission, but with still several concerns. Nevertheless, it may lead to interesting discussion at MIDL.